# BNT162b2 mRNA COVID-19 Vaccine Effectiveness in Patients with Coeliac Disease Autoimmunity: Real-World Data from Mass Vaccination Campaign

**DOI:** 10.3390/v15091968

**Published:** 2023-09-21

**Authors:** Amir Ben-Tov, Benjamin Lebwohl, Tamar Banon, Gabriel Chodick, Revital Kariv, Amit Assa, Sivan Gazit, Tal Patalon

**Affiliations:** 1Kahn Sagol Maccabi (KSM) Research and Innovation Center, Maccabi Healthcare Services, Tel Aviv 68125, Israel; 2Sackler Faculty of Medicine, Tel Aviv University, Tel Aviv 68125, Israel; 3Department of Medicine, Columbia University Irving Medical Center, New York, NY 10032, USA; 4Department of Epidemiology, Mailman School of Public Health, Columbia University, New York, NY 10032, USA; 5Department of Gastroenterology, Tel Aviv Sourasky Medical Center, Tel Aviv 68125, Israel; 6The Juliet Keidan Institute of Pediatric Gastroenterology and Nutrition, Shaare Zedek Medical Center, The Hebrew University, Jerusalem 91905, Israel

**Keywords:** celiac disease, COVID-19, immunization

## Abstract

Background: Data on COVID-19 vaccine effectiveness among patients with coeliac disease are currently lacking because patients with immune conditions were excluded from clinical trials. We used our coeliac disease autoimmunity (CDA) cohort to explore the effectiveness of the BNT162b2 mRNA COVID-19 vaccine in preventing SARS-CoV-2 infection among patients with CDA. Methods: This retrospective cohort study included patients with positive autoantibodies against tissue transglutaminase (tTG-IgA). In the primary analysis, the cohort included CDA patients who received two vaccine doses against COVID-19 and matched patients in a 1:3 ratio. Patients were divided into subgroups based on their positive tTG-IgA level at diagnosis and their current serology status. Results: The cohort included 5381 vaccinated patients with CDA and 14,939 matched vaccinated patients. The risk for breakthrough SARS-CoV-2 infection evaluated with Kaplan–Meier survival analysis via log-rank tests was similar between groups (*p* = 0.71). In a Cox regression survival analysis, the hazard ratio for breakthrough infection among patients with CDA compared to matched patients was 0.91 (95% confidence interval = 0.77–1.09). Conclusions: COVID-19 vaccination is effective in patients with coeliac disease autoimmunity. Vaccine effectiveness was comparable to the reference population.

## 1. Introduction

Severe acute respiratory syndrome coronavirus-2 (SARS-CoV-2) has caused over 300 million confirmed cases of COVID-19 globally and more than 5 million deaths as of January 2022 [1]. COVID-19 vaccines, including the BNT162b2 mRNA COVID-19 vaccine (Pfizer-BioNTech), have demonstrated excellent efficacy in preventing severe COVID-19 in phase III placebo-controlled randomized clinical trials and in real-world data analyses [2,3,4].

Coeliac disease (CD) and CD autoimmunity (CDA) are immune-mediated conditions characterized by autoantibodies to tissue transglutaminase, affecting around 1% of the population worldwide with trends towards an increase in incidence in recent years [5,6,7]. Patients with CDA were defined as having any tissue transglutaminase Immunoglobulin A (tTG-IgA) values above normal, as previously described [7]. Both conditions can manifest with or without symptoms, while CD diagnosis requires small intestinal enteropathy or a high level of antibodies.

CD has been associated with a decreased immune response to vaccination against the hepatitis B virus [8]. Studies o immunogenicity in other vaccines have shown comparable immune response to the general population, although the number of participants in those trials was low [9]. Patients with CD are not considered immune-suppressed, aside from those with severe refractory CD, though functional hyposplenism has been associated with CD, which likely accounts for a modestly increased risk of severe pneumococcal infection [10]. Patients with chronic diseases such as CD have concerns regarding their chances of severe and/or long COVID-19 and the effectiveness of vaccination [11], though the risk of severe COVID-19 does not appear to be increased in CD [12,13]. Since patients with immune conditions (including CD) were excluded from COVID-19 vaccine clinical trials [2], it is important to describe accumulating real-world data on vaccine effectiveness.

Few studies have addressed the effect of the SARS-CoV-2 vaccine in patients with celiac disease [14]. Studies from Norway and Italy showed that patients with CD had very good antibody response to vaccination that was comparable to healthy controls [15,16]. In contrast, a study from Jordan showed that patients with CD had lower humoral response than their matched controls [17]. However, none of these studies looked at the real-world performance of SARS-CoV-2 vaccines at the population level in patients with CD.

In the present population-based controlled study, we aimed to explore the effectiveness of COVID-19 vaccination in preventing SARS-CoV-2 infection in patients with CDA, both for newly diagnosed patients and patients with an established diagnosis with and without serologic remission.

## 2. Materials and Methods

### 2.1. Data Source

We used the Maccabi Healthcare Services (MHS) comprehensive database, a 2.5-million-member state-mandated health fund in Israel. MHS is the second largest of the four nationwide health funds in Israel, of which citizens can choose their respective healthcare memberships. The Maccabi health fund members represent 26.7% of the population and share similar socio-demographic characteristics with the overall Israeli population. The fund has maintained a computerized database of electronic health records since 1993, containing extensive longitudinal data on a stable population (with an approximate 1% annual turnover).

Additionally, the health fund has developed several computerized registries of major clinical conditions. These registries are continuously updated and detect relevant patients using pre-defined criteria (relying on coded diagnoses, extensive laboratory data, treatments, administrative billing codes, and active reporting by physicians).

### 2.2. Study Population and Design

This retrospective cohort study included vaccinated patients age ≥ 12 years with CDA with a minimum of two BNT162b2 mRNA COVID-19 vaccine doses. Only patients registered in the MHS database for at least 12 months prior to the first vaccine dose (index date) were included. Patients with CDA were further stratified according to the degree of tTG IgA elevation, as described below.

Individual matching was performed with a 1:3 ratio (patients with CDA to matched patients). Only patients without a record of tTG-IgA testing were considered for controls, and patients were matched based on birth year, sex, MHS branch (measured within a small geographic area across Israel), socioeconomic status (SES), and month of first vaccine dose. The SES was computed from patients’ residential areas and presented from low to high with a ranking system (1 as the lowest and 10 as the highest). 

Patients with a record of a prior positive polymerase chain reaction (PCR) result or those with a diagnosis of SARS-CoV-2 prior to the index date were excluded from this study. All patients were required to have at least two doses of the Pfizer vaccine with a minimum of 14 days follow-up after the second vaccination dose to ensure full vaccine effectiveness. The first vaccination date was used as the index date for this study for both those with CDA and their matched patients. 

PCR breakthrough infections post-second vaccination were recorded from 1 February 2021 until end of follow-up on 18 September 2021. 

### 2.3. Study Variables and Definitions

Patients in this study with CDA were stratified into four groups, compared to their matched patients, and then categorized as follows:Patients with CDA with an elevated tTG-IgA above ten times the upper limit of normal, described as patients with near-certain coeliac disease. This definition is based on the high positive predictive value of very high tTG-IgA levels [18].Patients with CDA with an elevated tTG-IgA followed by a subsequent normal tTG-IgA within two years pre-index date, where the subsequent normal test is at least one year since the first, described as patients with likely well-controlled coeliac disease.Patients with CDA with an elevated tTG-IgA followed by a subsequently elevated tTG-IgA within two years pre-index date, where the subsequent elevated test is at least one year since the first, described as patients with possibly not well-controlled coeliac disease.Newly diagnosed patients with CDA (a first elevated tTG-IgA test within a year from index date), described as patients with recently diagnosed coeliac disease autoimmunity.

Baseline characteristics described in this study include the matching parameters, such as age at index date, sex, SES, and the month of the first vaccine dose. Comorbidities at baseline from the MHS registries were also recorded, including diabetes, hypertension, cancer, cardiovascular disease (CARDIOVASCULAR DISEASE), chronic kidney disease (CKD), and immunosuppressed patients [19,20,21]. The immunosuppression registry consists of patients who were defined by an algorithm with inclusion criteria based on medication purchases and recorded diagnoses (International classification of diseases, ninth revision (ICD-9)) from an MHS physician. Body mass index (BMI) was also described and compared with patients’ most recent height and weight values pre-index date. Patients were considered to have underweight if their BMI was below 18.5, normal weight with values between 18.5 and 24.9, overweight with values between 25 and 29.9, and obesity if their BMI was above 30.

### 2.4. Study Outcome

The primary outcome was to compare vaccine effectiveness against SARS-CoV-2 infection (regardless of symptoms) in both CDA patients and controls. Cases were defined by at least 1 positive SARS-CoV-2 polymerase chain reaction (PCR) test recorded in the MHS computerized database. All such testing in MHS members is recorded centrally.

### 2.5. Statistical Analyses

Baseline characteristics were evaluated and comparative analyses were performed, namely, descriptive statistics presented as mean values (standard deviation (±SD)) and frequencies per parameter between coeliac and matched patients. The Mann–Whitney test was used to evaluate age and Pearson’s chi-squared test was used for discrete variables such as sex, SES, vaccination month, and comorbidities. 

Kaplan–Meier survival analyses were performed to evaluate time to breakthrough infection with SARS-CoV-2, defined as a positive PCR result 14 days post second vaccination date. Patients with CDA and their matched patients were compared via the log-rank test. Multivariate Cox regression analyses were also conducted where the independent variable was a PCR outcome event. The covariates entered in the models include comorbidities listed in Table 1 and body mass index (BMI) presented and stratified as normal, underweight, overweight, and obese based on patients’ last height and weight record within five years prior to the index date. Additionally, further comparisons were performed to assess the various categories of CDA patients, where Kaplan–Meier survival analyses and Cox regression analyses were applied for each. 

## 3. Results

Primary analysis—whole cohort

In this study, 7083 patients with CDA were identified in the MHS database with two confirmed COVID-19 vaccination doses and no prior record of a positive PCR test. After matching, there were 5381 patients with CDA matched to 14,939 patients without any record of tTG-IgA testing. The mean age in years for the CDA and matched patients was 33.38 (standard deviation (SD) ± 17.78) and 33.93 (SD ± 17.68), respectively. There were 3476 (64.6%) females in the whole cohort.

The CDA and matched patients included in the study predominantly received their first vaccination in December 2020 (37.1% and 37.5%) or January 2021 (31.1% and 31.8%). SES was also similar in both groups, where most patients were of high socioeconomic status (64.3% and 64.9%). Among the whole CDA cohort, the mean time from the first positive tTG-IgA was 6.73 years (SD ± 4.42). 

The patients with CDA and matched patients differed in certain comorbidities, where we observed differences in patients with diabetes, hypertension, cancer, cardiovascular disease, and immunosuppression (*p* < 0.05). Additionally, BMI differed between groups such that there were more overweight individuals in the CDA group (53.1%) compared to the matched patients (44.5%).

Vaccine effectiveness was evaluated with Kaplan–Meier survival analysis among the whole cohort (CDA n = 5381 and matched patients n = 14,939) to evaluate time to breakthrough SARS-CoV-2 infection, where there was no observed statistical difference between groups (*p* = 0.71, Figure 1A). A Cox regression survival analysis was performed for the cohort, resulting in a hazard ratio for breakthrough infection among CDA patients of 0.92 (95% confidence interval (CI) = 0.77–1.09). Stratification for different weight groups did not show significant differences between the groups. We had information on BMI level for 4857/5381 patients (90.02%). Compared to patients with a normal BMI, the hazard ratio for patients who were underweight was 1.29 (95% CI = 0.98–1.07), patients who were overweight had a hazard ratio of 0.95 (95% CI = 0.79–1.15), and patients with obesity had a hazard ratio of 0.77 (95% 0.59–1.00). See Table 2 for a detailed description of all covariates. 

The rate of hospitalization among both patients with CDA and their matched controls was very low, with no significant statistical difference. No deaths were recorded. 

Subgroup analysis

The same models were applied to the subgroups, and similar results were obtained. Vaccine effectiveness using Kaplan–Meier survival analysis for patients with likely well-controlled coeliac disease (n = 3190) and their matched patients (n = 8923) is presented in Figure 1B. There was no difference between groups in time to breakthrough infection (*p* = 0.38). The Cox regression survival analysis (Appendix A) demonstrated a hazard ratio for breakthrough infection of 0.9 (95% CI = 0.71–1.13). Patients with near-certain coeliac disease (n = 2310) and their matched patients (n = 6423) had no difference in time to breakthrough infection (Figure 1D, *p* = 0.29). The Cox regression survival analysis (Appendix A) demonstrated a hazard ratio for breakthrough infection of 0.81 (95% CI = 0.62–1.07) for patients with near-certain coeliac disease. Patients with likely not well-controlled CDA (n = 162) and their matched patients (n = 458) had no difference in time to breakthrough infection (Figure 1C, *p* = 0.3). The Cox regression survival analysis (Appendix A) demonstrated a hazard ratio for breakthrough infection of 0.57 (95% CI = 0.19–1.71). Patients newly diagnosed with CDA (n = 162) and their matched patients (n = 458) had no difference in time to breakthrough infection (Figure 1E, *p* = 098). The Cox regression survival analysis (Appendix A) demonstrated a hazard ratio for breakthrough infection of 0.99 (95% CI = 0.51–1.93).

## 4. Discussion

In this population-based study of all patients with at least two doses of COVID-19 vaccination with CDA in our health organization, we found that vaccine effectiveness is similar in CDA patients compared to matched patients. The effectiveness was evaluated across several subgroups of CDA patients, including newly diagnosed patients, and was similar in all groups. Patients with tTG-IgA above ten times the normal upper limit represent patients with near-coeliac disease as this cut-off represents a very high positive predictive value for villous atrophy and, in the proper clinical setting, is considered high enough for diagnosis without a biopsy in children and possibly in adults [22,23,24,25]. The similar effectiveness in these patients and in newly diagnosed patients is encouraging.

To the best of our knowledge, there are no published studies to date regarding vaccine effectiveness for COVID-19 in coeliac disease patients. Vaccine effectiveness in the general population was high in clinical trials and in real-world settings in several different statistical models. In patients with specific comorbidities, the results are conflicting and are related to the comorbidity and immune-modifying drugs in use. Dagan et al. showed slightly decreased vaccine effectiveness in patients with multiple coexisting conditions such as hypertension and diabetes [26]. Chodick et al. showed decreased vaccine effectiveness in immunosuppressed patients (71% vs. 90%) [4]. Real-world studies on patients with inflammatory bowel disease had encouraging results and showed similar effectiveness to matched control patients [27,28]. Ferri et al. demonstrated a lower seroconversion rate post-vaccine in patients with autoimmune systemic diseases. However, their results were based on serologic response and not on real-life data of infection rates [29]. Additionally, most of their patients were treated with immune-modifying treatments.

The fact that we did not observe any difference in vaccine effectiveness despite a significantly higher proportion of patients with comorbidities and immunosuppression in the CDA group further strengthens our findings that CDA is not associated with decreased effectiveness of mRNA COVID-19 vaccines. 

Our findings are supported by two studies that looked at the humoral response in patients with celiac disease. In the study by Ibsen et al. [15], there were 112 samples collected from celiac disease patients following the second dose of the vaccine schedule (Chadox1, Comirnaty, or Spikevax). Antibody levels overlapped with those of healthy controls. They concluded that the vaccine is effective and that patients with coeliac should follow the same vaccination routine as the general population. The study by Scalvini et al. [16] measured antibodies in patients with coeliac disease at three, six, and nine months following SARS-CoV-2 vaccination. Nine months following vaccination, all coeliac disease patients had an adequate humoral response that was comparable to healthy controls. 

The main strength of our study is our validated CDA cohort based on tTG-IgA performed in one national central laboratory and our centralized database on COVID-19 vaccination and breakthrough infections, allowing us to follow this large cohort accurately over time. In addition, the extended follow-up that lasted eight months included the third and fourth waves of the pandemic in Israel, dominated by the Alpha and Delta variants, respectively [30,31], which enabled analyzing the possible difference in waning immunity over time in patients with CDA, which did not occur. This study was conducted in a time period during the pandemic when almost all patients had COVID-19 tests due to their high availability and governmental requirement, therefore setting up a unique opportunity to explore the effects of mass population immunization.

This study has limitations relating to our structured database, which lacks information on pathological reports. Therefore, we focused on coeliac disease autoimmunity rather than coeliac disease. While acknowledging that CDA is not necessarily coeliac disease, it does represent the broad spectrum of autoimmunity to gluten peptides. This study was conducted prior to the Omicron surge and therefore cannot conclude on ongoing immunity against the Omicron variant. Patients with CDA differed from matched patients in BMI, diabetes, hypertension, and cancer proportion. These characteristics are risk factors for more severe COVID-19 but not for breakthrough infections. 

## 5. Conclusions

In conclusion, to the best of our knowledge, this is one of the first reports of real-world COVID-19 vaccine effectiveness in patients with CDA. The overall vaccine effectiveness was excellent and comparable to the reference population. Future studies should focus on long-term vaccine effectiveness and safety over time in patients with coeliac disease.

## Figures and Tables

**Figure 1 viruses-15-01968-f001:**
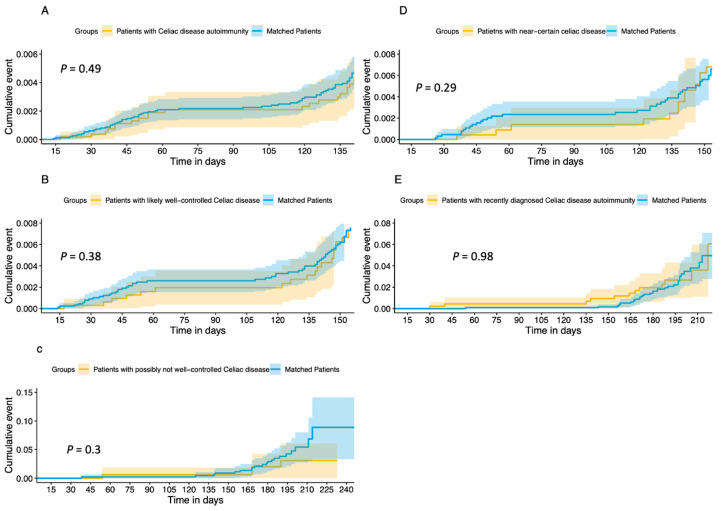
Cumulative incidence curves of Kaplan–Meier survival analysis for time-to-PCR-positive event for (**A**) patients with (**A**) Coeliac disease autoimmunity and their matched patients, (**B**) patients with likely well-controlled Coeliac disease and their matched patients, (**C**) patients with possibly not well-controlled Coeliac disease and their matched patients, (**D**) patients with near-certain celiac disease and their matched patients, and (**E**) patients with recently diagnosed Coeliac disease autoimmunity and their matched patients.

**Table 1 viruses-15-01968-t001:** Baseline characteristics (counts and %) for patients with coeliac disease autoimmunity and matched patients with comparison (*p*-value).

Baseline Patient Characteristics		Patients				*p*-Value
		Coeliac Disease Autoimmunity (n = 5381)		Matched (14939)		
Age (years) Mean (SD)		33.38	17.78	33.93	17.68	0.69
Sex	Male (N, %)	1905	35.4%	5349	35.8%	0.59
	Female (N, %)	3476	64.6%	9590	64.2%	
First vaccination month	Dec 2020	1996	37.1%	5599	37.5%	0.31
	Jan 2021	1675	31.1%	4755	31.8%	
	Feb 2021	224	4.2%	603	4.0%	
	Mar 2021	44	0.8%	86	0.6%	
	Apr 2021	8	0.1%	22	0.1%	
	May 2021	498	9.3%	1387	9.3%	
	Jun 2021	356	6.6%	1018	6.8%	
	Jul 2021	580	10.8%	1469	9.8%	
SES	Low (1–4)	610	11.3%	1723	11.5%	0.45
	Med (5–6)	1313	24.4%	3519	23.6%	
	High (7–10)	3458	64.3%	9697	64.9%	
Comorbidities	Diabetes	295	5.5%	559	3.7%	<0.001
	Hypertension	405	7.5%	1256	8.4%	0.04
	Cancer	205	3.8%	525	3.5%	0.32
	Cardiovascular diseases	235	4.4%	276	1.8%	<0.001
	Chronic kidney disease	206	3.8%	629	4.2%	0.23
	Immunosuppression	191	3.5%	383	2.6%	<0.001
BMI	Underweight (<18.5)	191	3.5%	383	2.6%	<0.001
	Normal weight (18.5–24.9)	813	15.1%	1812	12.1%	
	Overweight (25–29.9)	2860	53.1%	6651	44.5%	
	Obesity (>30)	1142	21.2%	3387	22.7%	
	Missing	524	9.7%	2102	14.1%	

**Table 2 viruses-15-01968-t002:** Cox regression analysis for patients with coeliac disease autoimmunity (n = 5381) and their matched patients—whole cohort (n = 14,939).

	*p*-Value	Hazard Ratio	95% Confidence Interval	
			Lower	Upper
Coeliac disease	0.33	0.92	0.77	1.09
BMI normal	Reference			
BMI underweight	0.07	1.29	0.98	1.70
BMI overweight	0.60	0.95	0.79	1.15
BMI obese	0.05	0.77	0.59	1.00
BMI missing	0.73	0.94	0.66	1.34
Diabetes	0.44	0.83	0.53	1.32
Hypertension	0.01	0.55	0.38	0.81
Cancer	0.03	0.53	0.30	0.92
Cardiovascular disease	0.65	1.13	0.67	1.88
Chronic kidney disease	0.14	0.68	0.41	1.13
Immunocompromised	0.19	0.67	0.37	1.22

## Data Availability

According to the Israel Ministry of Health regulations, individual-level data cannot be shared openly. Specific requests for remote access to deidentified community-level data to or to the code used for data analysis should be referred to the Kahn Sagol Maccabi Research & Innovation Center, MHS.

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
