# Peer review of "BNT162b2 mRNA COVID-19 Vaccine Effectiveness in Patients with Coeliac Disease Autoimmunity: Real-World Data from Mass Vaccination Campaign"

_viruses, 2023, doi:10.3390/v15091968_

Round 1

Reviewer 1 Report

This manuscript describes a retrospective study, in which authors aims to assess the effectiveness of the Pfizer-BioNTech vaccine against Covid-19 in patients with celiac disease. By using a large, population based healthcare database that reproduce characteristics of population in Israel, the authors compared the effectiveness of at least to doses of vaccine in adolescent and adults with that for the general population, by matching celiac patients with control subjects according to relevant variables. Patients were classified according to serum anti-tGA IgA values. Time to breakthrough infection with SARS-CoV-2, defined as a positive PCR result 14 days post second vaccination date was the primary endpoint of vaccine efficacy for this study, and multivariate models were applied to control all relevant variables with potential influence in the risk of developing Covid-19.

The authors found no differences in the time to breakthrough SARS-CoV-2 infection in the overall cohort of celiac patients, and also when subgroups analysis were performed, indication that vaccine effectiveness is similar in celiac patients, independently of the degree of control of the disease, compared to matched control subjects.

The topic addressed in this paper is novel in literature and the results are reliable, as seems to have been obtained by applying, apparently, appropriate methods.

My suggestions for the authors are aimed at improving the presentation of their results and the unequivocal interpretation of their data:

1. A new table 1 should be added describing on the clinical and demographical characteristics of the population, with include basic details as sex, age, time from celiac disease diagnosis, etc, and also detailed information of all the variables you used to match your cohorts of patients and controls. This table will be essential to understand baseline characteristics of your cohort and to evaluate potential external validity of your results.

2. Please indicate how many patients in each study cohort developed SARS-Cov-2 infection by positive PCR, how many developed Covid-19 and how many were admitted to hospital due to it. I understand that the difference was not significant, but raw data must be provided. This information could be summarized in the test and presented in detail in a new table.  

3. Table 1. Please indicate clearly in table 1 the variables and measures you are showing there. As an example, in the Age (years) line, the values represented are likely mean and standard deviation, but nothing is said about that. So please consider to indicate “Age (years), mean (SD)”. Please clarify in line 144 that values correspond to mean (±standard deviation). Also, in the table, please change “Count / Column N %” headings into “number of patients / %)”

4. In table 1, please include the numerical values of BMI that defined patients as underweight, overweight, etc. This could be also added to the method’s section of the manuscript.

5. Please use the same style for citing numbers in all your manuscript text and tables: Sometimes decimal values appears as “0.004” and sometimes as “ .56”. This should be unified in a single notation.

6. Tables should be self-explanatory: In tables 2 to 6, please consider to annotate “Hazard ratio”, instead of HR, and “95% confidence interval”, instead of 95% CI.

7. Tables 2 to 6: Please complete the concept of “Cardio” in the content, and clarify CKD abbreviation.

8. Consider to move tables 2 to 6 to supplementary material.

Reviewer 2 Report

The authors study the effectiveness of vaccination against COVID-19 in a population of patients older than 12 years of age, who have had or have positive anti-transglutaminase IgA antibodies (Celiac Disease Autoimmunity, CDA), compared to controls matched for age and sex ( who have never had an anti-transglutaminiase antibody test). We assume that the study group is representative of the population of celiac patients, who are stratified into 4 groups in this study. We can interpret the control population as a non-celiac population (but not a healthy population). The study therefore follows a case-control design. Both groups were vaccinated with at least two doses of the Pfizer-BioNTech RNA vaccine. Follow-up is carried out for up to 8 months post-vaccination and the declaration of infection by SARS-Cov2 defined as positivity for the specific PCR is recorded: this fact has being considered the independent output variable. The evolution of each group is compared using Kaplan-Meier survival analysis and Cox multivariate analysis, with other clinical variables collected.

The analyzed casuistic is considerable, and the design, as a whole, can be considered correct, so the results are credible and the conclusions representative: individuals with celiac disease autoimmunity present the same post-vaccination infection rate as the rest of the vaccinated population as a whole. However, there are some issues that, perhaps, could be discussed: -I do not know if in Israel the epidemiological control of the pandemic was so rigorous as to assume that all those infected declared themselves and a PCR was performed on them. It is likely that there were mild diseased unrecorded cases. Are we to assume that the two groups had the same proportion of unrecorded cases? -There is no doubt that most of the weight in avoiding infections was due to vaccination, but also to the implementation of epidemiological and behavioral measures in the population. These measures were all the more stringent the more real or perceived risk each individual had. Can we be sure that this was also matched? -As an immunologist, I still do not have data on whether the immune response of each group to the vaccine was similar: it was in terms of the infection rate, but there is no data on the severity of the infections in each group. (I really miss that data, such as the levels of anti-S1 antibodies after vaccination and their persistence, the presence of memory cell immunity, or the response with anti-N antibodies after infections. Perhaps these ones are not related with the Real World). -I can understand the rationale for the “well controlled”, “not well controlled” and “recently diagnosed” CDA groups, but I have a hard time seeing the rationale for the “near-certain Celiac Disease” group. The reason for this group should be explained. -In figure 1, curve c ("well controlled" CDA group), a difference is observed between the groups after 6 months of evolution. Can there be any explanation?

Reviewer 3 Report

Could the authors be so kind and be more specific on the end-points of the study, when talking about vaccine efficacy.

For patients with comorbidities included in the study these comorbidities were prior to CD or not

Did the author see any change in VE in regard of the BMI of their patients

Round 2

Reviewer 1 Report

The authors have provided prompt responses to all my criticisms of the previous version. I believe that the manuscript has gained clarity

Author Response

Thank you for all your comments